# Conservative Treatment for Spontaneous Resolution of Postoperative Symptomatic Thoracic Spinal Epidural Hematoma—A Case Report

**DOI:** 10.3390/medicina59091590

**Published:** 2023-09-02

**Authors:** Stjepan Dokuzović, Mario Španić, Sathish Muthu, Jure Pavešić, Stjepan Ivandić, Gregor Eder, Bogdan Bošnjak, Ksenija Prodan, Zoran Lončar, Stipe Ćorluka

**Affiliations:** 1Spinal Surgery Division, Department of Traumatology, University Hospital Center Sestre Milosrdnice, 10000 Zagreb, Croatia; stjepan.dokuzovic@kbcsm.hr (S.D.); jure.pavesic@kbcsm.hr (J.P.); 2Akromion Special Hospital for Orthopaedic Surgery, 49217 Krapinske Toplice, Croatia; mario.spanic@hotmail.com; 3Orthopaedic Research Group, Coimbatore 641045, Tamil Nadu, India; drsathishmuthu@gmail.com; 4Department of Biotechnology, Faculty of Engineering, Karpagam Academy of Higher Education, Coimbatore 641021, Tamil Nadu, India; 5Department of Orthopaedics, Government Medical College, Karur 639004, Tamil Nadu, India; 6Traumatology Department, University Hospital Centre Sestre Milosrdnice, 10000 Zagreb, Croatia; stjepan.ivandic@kbcsm.hr (S.I.); gregor.eder@kbcsm.hr (G.E.); 7General Hospital, Croatian Veterans, 49210 Zabok, Croatia; bosnjak.bogdan@gmail.com; 8Clinical Department of Diagnostic and Interventional Radiology, Department of Traumatology, University Hospital Center Sestre Milosrdnice, 10000 Zagreb, Croatia; ksenija.prodan@kbcsm.hr; 9Anesthesiology, Intensive Care and Pain Management Division, Traumatology Department, University Hospital Centre Sestre Milosrdnice, 10000 Zagreb, Croatia; zoran.loncar@kbcsm.hr; 10St. Catherine Specialty Hospital, 10000 Zagreb, Croatia; 11Department of Anatomy and Physiology, University of Applied Health Sciences, 10000 Zagreb, Croatia

**Keywords:** thoracic, spine, trauma, fracture, hematoma

## Abstract

*Introduction*: Postoperative epidural hematomas of the cervical and thoracic spine can pose a great risk of rapid neurological impairment and sometimes require immediate decompressive surgery. *Case Report*: We present the case of a young patient operated on for stabilization of a two-level thoracic vertebra fracture who developed total paralysis due to an epidural hematoma postoperatively. The course of epidural hematoma was quickly reversed with the help of a conservative technique that prevented revision surgery. The patient regained complete neurologic function very rapidly, and has been well on every follow-up to date. *Conclusion*: There is a role of similar maneuvers as described in this case to be employed in the management of postoperative epidural hematomas. However, prolonged watchful waiting should still be discouraged, and patients should remain ready for revision surgery if there are no early signs of rapid recovery.

## 1. Introduction

An epidural hematoma can rapidly lead to neurological impairment [1]. Even though spinal surgeries are the most common cause, epidural hematomas can also arise spontaneously due to injury, blood thinning medication, bleeding from abnormal blood vessels in the spinal cord (such as an arteriovenous malformation), epidural steroid injections, or spinal or epidural anesthesia [2,3,4]. Given their potentially disastrous long-term effects, epidural hematomas warrant the attention of all spinal surgeons.

Epidural hematomas, although rare, can be a severe postoperative complication after spinal surgery. The reported incidence of a symptomatic epidural hematoma is less than 1% ranging from 0.10% to 0.69% [5,6]. While asymptomatic epidural hematomas are relatively more common postoperatively, symptomatic epidural hematoma mandates immediate attention and appropriate management to avoid untoward long-term complications. Various studies looking into the pharmacological prophylaxis for deep vein thrombosis and pulmonary embolism have reported epidural hematoma rates of less than 1% [7]. Although commonly employed, wound drains cannot be fully relied upon to eliminate the occurrence of epidural hematomas [8].

The diagnosis of symptomatic postoperative epidural hematoma necessitates a high index of suspicion and is made based on clinical symptoms such as evolving sudden postoperative axial pain in the area of surgery with neurological deficit and urinary retention [9]. In such cases, an emergency magnetic resonance imaging (MRI) is carried out to confirm the diagnosis, followed by emergency evacuation of the hematoma within 6–12 h [10]. Here, we present a case of post-operative epidural hematoma that was quickly reversed with the help of a conservative technique that prevented revision surgery.

## 2. Case Report

A 36-year-old male patient was admitted to our emergency trauma department with compressive Th6 and Th12 fractures, as shown in Figure 1 sustained from a fall on the head and upper spine. The initial examination ruled out accompanying head or any organ injury. There were no rib fractures or injuries to the extremities. The patient was neurologically normal upon admission.

There were no fracture lines through the posterior elements or indirect signs of rupture of the posterior tension band. Low-molecular-weight heparin was routinely applied the evening before surgery. Spinal stabilization surgery was performed the day after admission after a brief period of preoperative preparation. Intraoperatively, no disturbance of the posterior tension band was observed (interspinous ligaments were intact, and the laminar bone showed no subtle fractures). All transpedicular screws were placed under fluoroscopic guidance in two planes, as shown in Figure 2. There was no breach of the medial pedicular walls, and no laminectomy was performed, since there was no need initially. Intraoperatively, there was minimal blood loss, and the anesthesiologist did not report any significant changes in blood pressure or pulse.

Upon awakening from general anesthesia, the initial postoperative neurological state was without any impairment, and the patient was able to urinate spontaneously. Approximately 45 min after surgery, he began to develop signs of hypesthesia in the lower extremities, weakness in the legs, an unsettling sensation in the abdomen, and was covered with cold sweat. Due to neurological worsening, an emergency MRI of the thoracic spine was performed, which revealed a hyperacute epidural hematoma that developed at the level of the Th6 fracture and spread cranially to Th4, as shown in Figure 3. The width was measured 7 mm, and compressed the spinal cord against the laminae, as shown in Figure 4.

Upon completion of the MRI, total paralysis was observed in the legs, as well as skin anesthesia up to the level of the lower ribcage (roughly corresponding to the Th7 dermatome). Emergency revision surgery was immediately planned, and the patient was placed in a semi-seated position while awaiting entry into the operating room. There was a roughly 45 min wait for the operating theatre to be ready, during which time rapid and complete resolution of the sensory and motor deficit was observed. The need for emergency surgery was revised and postponed due to the observation of spontaneous neurological recovery.

Serial neurological exams were conducted every 30 min, during which there was no worsening of neurological status. The daily dose of low-molecular-weight heparin was skipped on the day of the incident to avoid any potential ongoing minor epidural bleeding, and the patient was encouraged to frequently test his muscle strength by isometric contractions of the groups of the leg muscle while recovering in bed.

Normal bladder function was restored 2 days after surgery. Repeat MRI evaluation was done on third postoperative day, demonstrating significant resolution of the hematoma, as shown in Figure 5 and Figure 6. The subsequent hospital stay was uneventful. The patient was mobilized the day after surgery, with pain medication tapered in 2 days; the wound healed primarily without signs of inflammation or dehiscence, and the patient is now fully independent in daily tasks that do not require much stress on the spine.

## 3. Discussion

A symptomatic epidural hematoma can occur post-surgery, since decompressive methods expose the dura to pressure from bleeding [11]. The bleeding can come from various places, such as epidural veins, muscle surfaces, or bone surfaces disrupted during fusion. Particularly, the Batson’s plexus, a vein network, can bleed profusely, especially during posterior interbody fusion procedures. Visualization of the bleeding source, hemostasis, and hematoma evacuation may be quite challenging to manage, especially in obese patients [12]. Small arteries near facet joints that are injured during surgical dissection may also contribute. Injuries in paraspinal blood vessels can become active post-retraction. It is crucial to achieve hemostasis throughout and reassess before wound closure. Addressing a significant bleed during the initial surgery is preferable to needing to return to the operative room due to hematoma-induced symptoms.

An epidural hematoma that presents with symptoms following spine surgery is an uncommon occurrence, yet it has the potential to cause irreversible nerve damage and lasting serious disability. Factors that increase the likelihood of developing an epidural hematoma are older age, the use of blood-thinning medication before or after surgery, or a procedure involving the removal of multiple laminae [13]. If clinical symptoms suggest a hematoma, an urgent MRI is essential. An MRI helps confirm the diagnosis, showing fluid causing thecal sac compression. While some compression is normal and accepted, severe compression demands attention. The differential diagnosis for postoperative weakness includes tumor, infection, epidural hematoma, and disc herniation. An MRI serves as a confirmatory test, and it is crucial to correlate its findings with patient symptoms.

Sokolowski and colleagues created a “critical ratio” for evaluating thecal sac compression severity, but it is not an absolute criterion to decide on the method of management [14]. After diagnosing a symptomatic epidural hematoma, immediate evacuation is necessary to prevent permanent ischemic damage to the spinal cord. The existing incision can be used or extended for access. Potential bleeding sources need to be identified and eliminated using techniques such as coagulation and bone wax. Subfascial drains are typically employed during wound closure. Coagulation profiles should be conducted, and anticoagulants are paused if used as in our case. The time delay to evacuation and pre-evacuation neurologic impairment are key factors in prognosis. Neurologic recovery is maximized when deficits occur gradually, are incomplete, and when evacuation is undertaken within the first 12 h of onset [15]. After diagnosis, prompt surgical intervention should be prioritized.

In the case presented, due to the spontaneous resolution of the hematoma over a short time period, emergency revision surgery was avoided. Emergency revision is no guarantee of neurological recovery, as the procedure may require extensive decompression, costotransversectomy, and slight cord manipulation to access the anteriorly located hematoma; these are maneuvers that can cause further injury to the fragile mid-thoracic spinal cord. The bleeding most likely occurred from the fracture line itself. The assumption is that the slight elevation of the upper part of the body to a semi-seated position of approximately 20° (initially conducted to avoid further cranial migration of the hematoma) allowed caudal diversion (under gravity alone) of the hyperacute hematoma to the lumbar epidural space, where there is significantly more space to accommodate the few milliliters of blood.

To date, literature on such simple spontaneous resolutions of an otherwise potentially devastating complication is scarce. The reported cases are mostly cervical spinal epidural hematomas with spontaneous resolution without any known identifiable reason for the spontaneous resolution [16,17]. To the authors’ knowledge, this is the first case report of the spontaneous resolution of a thoracic postoperative spinal epidural hematoma with a simple conservative measure following thoracic stabilization surgery.

Continued passive observation for an extended period is not recommended, and patients should be prepared for the possibility of a follow-up surgical intervention if early indications of swift recovery are not apparent. This is to suggest that while it is crucial to give the body some time to heal and show signs of improvement after initial surgery, an excessively long period of inaction or “watchful waiting” may not be in the patient’s best interest. If the patient does not show signs of rapid recovery in the initial stages following the procedure, they should be mentally and physically prepared for a potential secondary surgery or revision surgery. This secondary surgery aims to rectify issues that have not been resolved with the initial operation.

## 4. Conclusions

Although in our case, this was sudden and most unexpected, there may be a rationale to attempt similar simple elevation maneuvers when spontaneous or postoperative epidural hematomas are encountered as a cause of acute neurological deterioration. Prolonged watchful waiting should still be discouraged, and patients should remain ready for revision surgery if there are no early signs of rapid recovery.

## Figures and Tables

**Figure 1 medicina-59-01590-f001:**
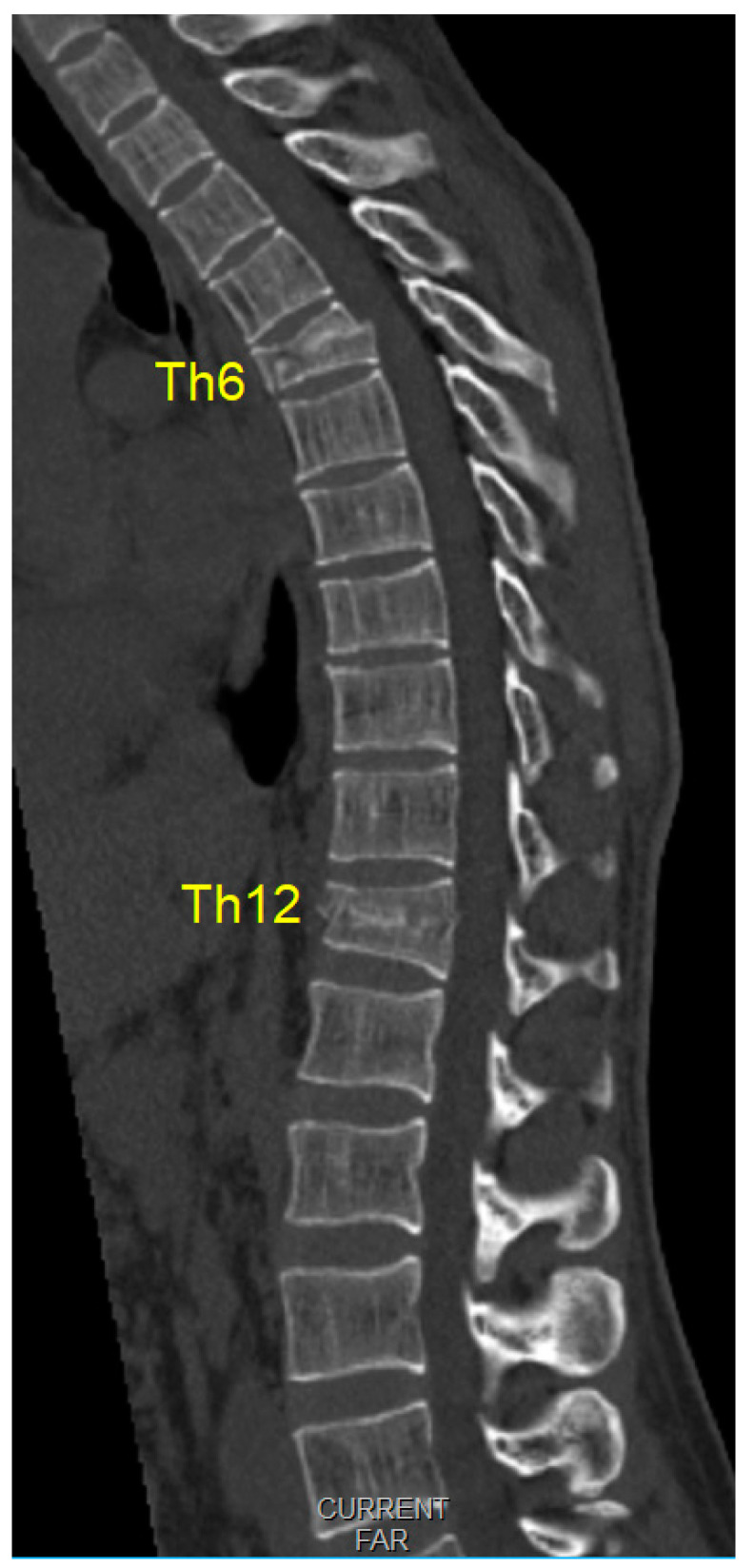
Initial multi-slice computerized tomography scans (midsagittal view) of compression fractures (Th6 A3 type fracture, and an Th12 A1 type fracture, classified according to the AO system).

**Figure 2 medicina-59-01590-f002:**
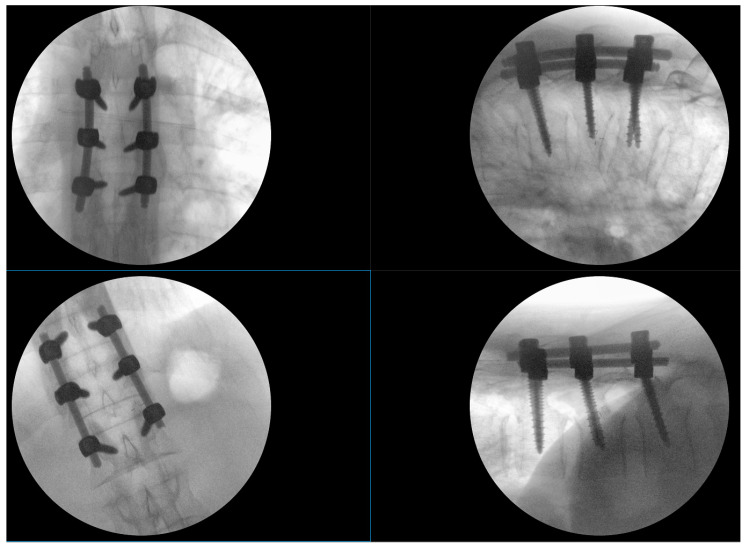
Intraoperative fluoroscopy images of the two short stabilizations performed; the upper images show the Th5–Th7 fixation that includes the fractured vertebra, and the lower images show the Th11–L1 fixation, the height of the fractured Th12 having been restored using postural reduction on the operating table.

**Figure 3 medicina-59-01590-f003:**
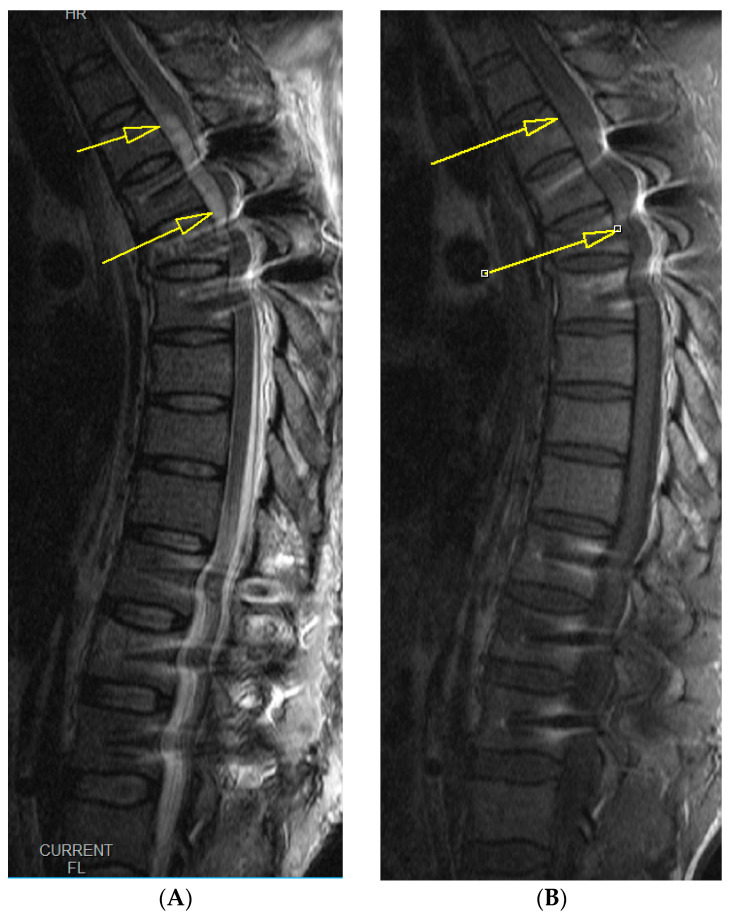
Emergency T2- (**A**) and T1 (**B**)-weighted MRI images, midsagittal views of the thoracic spine, show a hyperacute hematoma developing anterior to the spinal cord and spreading cranially to Th4, measuring 7 mm at its thickest. The yellow arrows in both the iamges depict the hematoma that is visible despite interference from implant artifacts.

**Figure 4 medicina-59-01590-f004:**
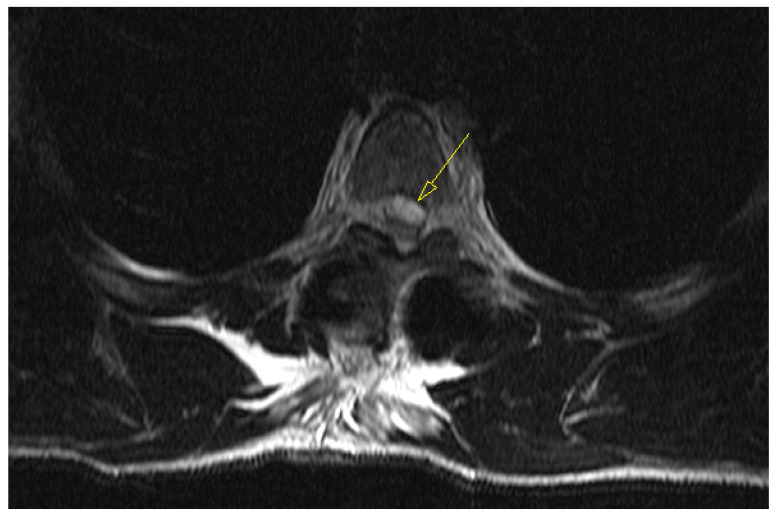
Emergency T2-weighted MRI image, an axial view at the Th5 level, revealing hyperacute hematoma development anterior to the spinal cord and causing significant dorsal displacement and compression.

**Figure 5 medicina-59-01590-f005:**
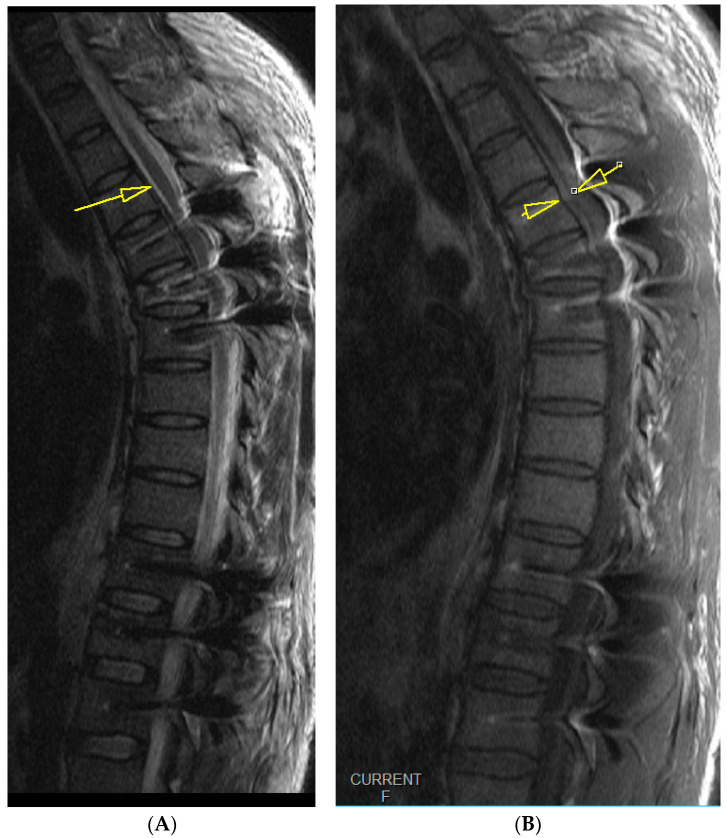
Follow-up midsagittal views of T2-weighted (**A**) and T1-weighted (**B**) MRI images taken 3 days after the initial emergency MRI showing near complete resolution of the hematoma, measuring 2 mm at its thickest.

**Figure 6 medicina-59-01590-f006:**
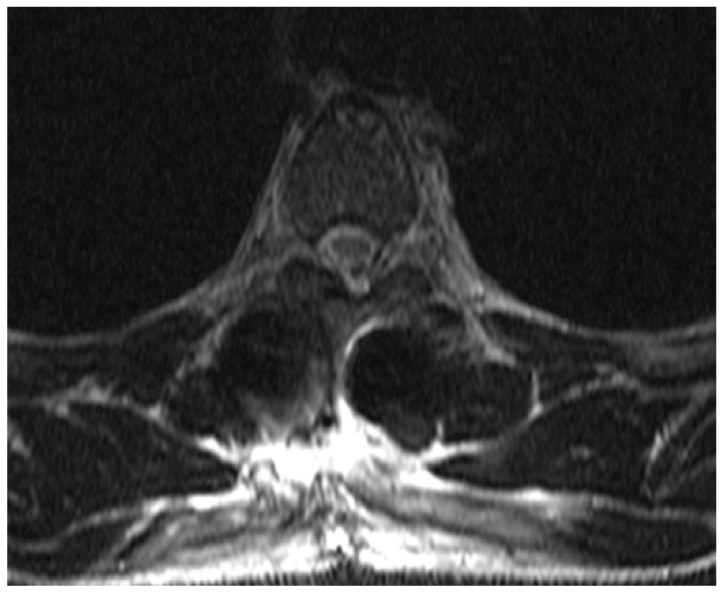
Follow-up axial view of an MRI taken 3 days after the initial emergency MRI, showing significant resolution of the hematoma. The view is at the level of Th4.

## Data Availability

No new data were created or analyzed in this study. Data sharing is not applicable to this article.

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
