# Peer review of "Conservative Treatment for Spontaneous Resolution of Postoperative Symptomatic Thoracic Spinal Epidural Hematoma—A Case Report"

_medicina, 2023, doi:10.3390/medicina59091590_

Round 1
Reviewer 1 Report
Abstract
Please reword – regained complete neurology as it does not make sense in English.
Introduction
No need to explain what epidural hematoma
Please restructure the sentance on drains as it currently has no sense in current structure.
Please reword to
In such cases an emergency magnetic resonance imaging (MRI) is done to confirm the diagnosis and if needed an emergency evacuation of the hematoma is performed within 6-12 hours.
We present a case instead the case
Case report
Is the information on soft surface really needed?
Figure 1 AO A3 and AO A1 Stands for? The figure captions have to be understandable on their own
Posterior tension band? Line 71
Line 75 Delete all
Line 77 delete primaliry..............initially.
Line 94 no need to report the milimeters as it is already done in fig 3
Line 95 Delete Most likely the bleeding occurred from the fracture itself. This sentence is suitable for discussion
Figure 3 – reword (implant artifacts obscuring the view somewhat)
Reword : During the roughly 45 minute waiting time (while another surgery was finishing up),
Fig 5 substitute significant with important
Line 129 to 137 are suitable for discussion
Discussion
when decompressive methods expose neural components – make understandable
Line 150 – what is difficult to manage?
Line 151 – use other word tha exposure
Line 153 before closing what? Also line 172
Substitute operation for surgery
159 better define age as now it is just open to interpretation- younger, older?
Lin 162 needs reference for CT usage
The time frame of 6 to 12 hrs is repeated too many times.
It would be interesting to support whay this time frame is important
Line 178 just the literature is scarce
181 delete in them
In the paragraph 178-184 please explain to the reader how ohter thoracic hematomas were treated in the literature. It would also be good to convey if there is any data in the literature on differences in occurence of hematomas cervical vs thoracic vs lumbar. Mostly in line179, others were lumbar?
Line 195 – why is there nr. 5?
Lin2 193-195 are obsolete
english should be improved
Reviewer 2 Report
Dear Editor and Authors,
As always a pleasure to be asked to review for Medicina this case report titled “Conservative Treatment for Spontaneous Resolution of Postoperative Symptomatic Thoracic Spinal Epidural Hematoma – A Case Report” by Dr. Dokuzović and colleagues from Croatia.
In this case report the authors describe a 36-year-old male patient with compressive Th6 and Th12 fractures sustained from a fall, who developed total paralysis due to an epidural hematoma postoperatively which spontaneously resolved and was thus managed conservatively.
This is an interesting case which had a good outcome for the patient. The authors highlight the mechanism they believe aided in the resolution (the body position of the patient) and this is a valuable message to the surgical community.
The report is well written and very nicely illustrated with accompanied radiological imagery. The language is clear and only requires some very minor English proofreading.
In conclusion, I enjoyed reading this report, it was easy and clear to me and I think it is worth while to be reported in the literature. My recommendation is acceptance. Thank you and take care all.
P.S. To Authors: Good luck, he (and you) were very lucky!
Very minor language editing required.
